# Diffuse Pulmonary Meningotheliomatosis: Clinic-Pathologic Entity or Indolent Metastasis from Meningioma (or Both)?

**DOI:** 10.3390/diagnostics13040802

**Published:** 2023-02-20

**Authors:** Laura Melocchi, Giulio Rossi, Mirca Valli, Maria Cecilia Mengoli, Michele Mondoni, Luigi Lazzari-Agli, Giacomo Santandrea, Fabio Davoli, Chiara Baldovini, Alberto Cavazza, Thomas V. Colby

**Affiliations:** 1Pathology Unit, Department of Oncology, Fondazione Poliambulanza Hospital Institute, 25124 Brescia, Italy; 2Operative Unit of Pathologic Anatomy, Ospedale Infermi, Azienda USL Romagna, 47900 Rimini, Italy; 3Operative Unit of Pathology, Azienda USL/IRCCS, 42123 Reggio Emilia, Italy; 4Respiratory Unit, ASST Santi Paolo e Carlo, Department of Health Sciences, Università degli Studi di Milano, 20142 Milan, Italy; 5Pulmonology Unit, Ospedale Infermi, Azienda USL Romagna, 47900 Rimini, Italy; 6Department of Thoracic Surgery, Azienda USL Romagna, S. Maria delle Croci Teaching Hospital, 48121 Ravenna, Italy; 7Cardiovascular Pathology Unit, Department of Pathology, IRCCS, St. Orsola Hospital, University of Bologna, 40138 Bologna, Italy; 8Department of Laboratory Medicine and Pathology (Emeritus), Mayo Clinic Arizona, Scottsdale, AZ 13400, USA

**Keywords:** meningotheliomatosis, lung, computed tomography, meningioma, transbronchial biopsy

## Abstract

Pulmonary minute meningothelial-like nodules (MMNs) are common incidental findings in surgical specimens, consisting of tiny proliferation (usually no larger than 5–6 mm) of bland-looking meningothelial cells showing a perivenular and interstitial distribution, sharing morphologic, ultrastructural, and immunohistochemical profiles with meningiomas. The identification of multiple bilateral MMNs leading to an interstitial lung disease characterized by diffuse and micronodular/miliariform patterns radiologically allows the diagnosis of diffuse pulmonary meningotheliomatosis (DPM). Nevertheless, the lung is the most common site of metastatic primary intracranial meningioma, and differential diagnosis with DPM may be impossible without clinic–radiologic integration. Herein, we report four cases (three females; mean age, 57.5 years) fitting the criteria of DPM, all incidentally discovered and histologically evidenced on transbronchial biopsy (2) and surgical resection (2). All cases showed immunohistochemical expression of epithelial membrane antigen (EMA), progesterone receptor, and CD56. Notably, three of these patients had a proven or radiologically suspected intracranial meningioma; in two cases, it was discovered before, and in one case, after the diagnosis of DPM. An extensive literature review (44 patients with DPM) revealed similar cases with imaging studies excluding intracranial meningioma in only 9% (4 of 44 cases studied). The diagnosis of DPM requires close correlation with the clinic–radiologic data since a subset of cases coexist with or follow a previously diagnosed intracranial meningioma and, thus, may represent incidental and indolent metastatic deposits of meningioma.

## 1. Introduction

Minute meningothelial-like nodules (MMNs) are common incidental findings of unclear significance encountered during a histologic examination of lung tissue, primarily surgical lung biopsies, generally lacking any clinical significance [1,2,3,4]. Their frequency depends on the extent of sampling in lung resections performed for other diseases, namely, primary/metastatic malignancies, infections, and interstitial lung diseases (ILD), among others [5,6,7,8]. When MMNs are widely disseminated in both lungs, they may be recognizable at chest CT scan leading to a diagnosis of interstitial lung disease (ILD) with the micronodular pattern reported in the literature as diffuse pulmonary meningotheliomatosis (DPM) [9,10,11,12,13,14,15,16,17,18,19,20,21,22,23,24,25,26,27,28,29,30,31]. Since MMNs are common incidental findings, the designation DPM presupposes a lack of identification of other lesions as a cause of micronodules.

At CT imaging, the differential diagnosis mainly includes a metastatic disease and a granulomatous process (e.g., sarcoidosis and infections) [10,11,12,13,14,15,16,17,18,19,20,21,22,23,24,25,26,27,28,29,30,31,32]. 

In light of the micronodular pattern with perivenular and interstitial distribution, DPM may be identifiable on a transbronchial biopsy, avoiding more invasive surgical procedures [27,28,29,33,34]. Indeed, at least four cases of DPM have been diagnosed by transbronchial biopsy, including one transbronchial cryobiopsy [11,27,28,29,33]. 

At histology, DPM is characterized by the tiny (generally up to 5–6 mm in maximum diameter) and irregular proliferation of bland-looking spindle-to-epithelioid MMNs, expressing meningothelial cell markers as epithelial membrane antigen (EMA), CD56, and progesterone receptor [5,6,7,8,9,10,11,12,13,14,15,16,17,18,19,20,21,22,23,24,25,26,27,28,29,30,31,32,33,34,35,36,37,38,39,40].

At the molecular level, few studies have investigated the nature of MMNs [41,42,43,44]. The most interesting studies have demonstrated that MMNs share neurofibromatosis (NF)-2 gene deletion with intracranial as well as primary pulmonary meningiomas, with the proposal that MMNs may be precursor neoplastic lesions [41,42,43,44]. On the other hand, clinic-pathologic presentations identical to DPM have been described in association with pulmonary meningioma and in patients with intracranial meningioma [26,45,46,47,48]. Interestingly, micronodular and cavitated/cystic lesions have been described in CT images in both DPM and pulmonary metastasis from intracranial meningioma [26,45,46,47,48].

Herein, we report four additional patients with micronodular interstitial lung disease at chest CT and histologic confirmation of MMNs (two from transbronchial biopsy, two on surgical resections), three of which had a previous (two cases) or concomitant history intracranial meningioma (one case), a finding that could challenge the definition of DPM as a primary pulmonary tumor entity of meningothelial cells. 

An exhaustive review of 44 cases of DPM previously appeared in the literature was performed to compare our results with those published in other reports in order to propose consistent diagnostic criteria for this intriguing pathology.

## 2. Materials and Methods

For this study, we identified 4 cases of DPM from the archival files of the Pathology Units of Infermi Hospital of Rimini (Italy), Santa Maria delle Croci Hospital of Ravenna (Italy), and Santa Maria Nuova Hospital of Reggio Emilia (Italy), and from the consultation files of the authors (from January 2008 to December 2021). Collected cases consist of consecutive biopsies without any selection bias, disclosed during routine practice or consultations. In addition, 25 pulmonary surgical resections from the above files (from January 2013 to December 2017), containing at least 3 MMNs and 20 fetal/neonatal autopsies (from January 2013 to December 2017), were also examined at immunohistochemistry to highlight and characterize MMNs. All cases (routine hematoxylin-eosin-stained slides and immunohistochemical stains) were reviewed at the multiheaded microscope by 3 pathologists (GR, MV, AC). 

Immunohistochemical studies were performed using the following prediluted primary antibodies (Ventana/Roche Medical Systems, Tucson, AZ, USA): CD10 (clone SP67), estrogen (clone SP1), and progesterone (clone 1E2) receptors, EMA (clone E29), HMB45 (clone HMB45), CD56 (clone 123C3), S100 (polyclonal), chromogranin (clone LH2K10), synaptophysin (clone MRQ49), pan-cytokeratins (clone AE1/AE3), TTF-1 (clone 8G7G3/1), desmin (clone DER-11), and smooth muscle actin/SMA (clone 1A4) in an automated immunostainer (ULTRA, Ventana/Roche Medical Systems, Tucson, AZ, USA). Appropriate positive internal and/or external controls were included in each batch. Clinical findings at the presentation and follow-up information were obtained from the medical records and the referring physicians. This study was conducted in accordance with the precepts of the Helsinki Declaration; all data were handled anonymously and in accordance with the local institutional ethical board protocols. A comprehensive search of the literature on PubMed/Medline variably crossing the words “meningotheliomatosis”, “meningothelial-like nodules”, “pulmonary“, “diffuse”, and “multiple” was performed until 31 January 2023, without language restrictions. 

## 3. Results

The clinical pathologic features of the cases included in the current series are summarized in Table 1.

The case series consists of three women and one man, with a mean age at diagnosis of 57.5 years (range 51–68 years). Three patients were never-smokers, and all four had an incidentally discovered diffuse micronodular interstitial lung disease (ILD) at a CT scan during imaging studies performed for other reasons. In two cases, the radiologic findings were considered in favor of sarcoidosis, while metastatic disease and diffuse idiopathic pulmonary neuroendocrine cell hyperplasia (DIPNECH) were originally proposed in the other two cases based on the imaging studies. In any case, the diagnosis of DPM was suspected by radiologists. The diagnosis of DPM was obtained from transbronchial biopsy, while two patients underwent surgical lung biopsy. Three patients had a past medical history of intracranial meningioma (two in the frontal and one in the temporal region) ranging from 4 to 10 mm in maximum diameter. All patients are alive and well (range, 38–99 months) without any specific therapy for DPM. In consideration of the presence of intracranial meningioma, three cases (cases #2, 3, and 4) of DPM were finally considered most consistent with indolent metastatic disease from meningioma, while a diagnosis of primary DPM was made in the last case (case #1). 

At the CT scan, all cases were quite similar, with numerous tiny nodules ranging from 1 to 6 mm in maximum diameter, randomly distributed prevalently at the peripheral zone of both lungs without lobar predominance, showing the rounded shape and solid or ground glass density. Some nodules showed central lucency (Figure 1A and Figure 2A).

Histologic demonstration of pulmonary meningothelial-like nodules was performed via transbronchial in two cases and surgical lung biopsies in two cases. 

The light microscope examination of the nodules showed a bland-looking proliferation of oval-to-spindle-shaped or epithelioid cells widening the interalveolar interstitial tissue, generally distributed around venules (Figure 1B,C and Figure 2D,E). Clusters of these neoplastic cells were vaguely arranged in whorl-like structures. Of note, central enlargement of alveolar airspaces surrounded by the proliferation of neoplastic cells seems to determine the cavitation/cystic changes centered in some nodules at CT scan (Figure 1A and Figure 2A). The cellular growth was often accompanied by variable amounts of collagenized fibrous tissue forming a scaffolding in both transbronchial and surgical samples. None of the four cases had any additional/alternative findings that could explain the formation of micronodules.

Immunohistochemically, the lesions showed consistent expression of epithelial membrane antigen (EMA), progesterone receptor, and CD56, whereas neuroendocrine, smooth muscle, melanocytic, vascular markers, CD10, cytokeratins, TTF1, and estrogen receptors were all negative (Figure 1D–F and Figure 2F,G). The proliferative index by Ki67 was low (~1%).

After a multidisciplinary discussion with clinicians and radiologists, a diagnosis of diffuse pulmonary meningotheliomatosis (DPM) was made. However, in cases #3 and #4 (Figure 2B,C), a prior diagnosis of intracranial meningioma was subsequently discovered in further review of the clinical records of the patients, and in a third (#2), an 8 mm diameter presumed that the meningioma of the right frontal lobe was disclosed at imaging studies performed after the diagnosis of DPM. A brain CT scan in case #1 did not evidence the intracranial pathologic lesions.

All four patients were still asymptomatic at follow-up (ranging from 38 to 99 months), and no therapy was administered for DPM.

A series of 25 surgical lung biopsies performed for other pathologies (16 primary lung cancers, 3 metastatic tumors, 3 ILD with usual interstitial pneumonia pattern, and 2 mycetomas) showed more than three MMNs, and, analyzed immunohistochemically, demonstrated the same profile of MMNs disclosed in DPM: all sporadic MMNs displayed expression of EMA, CD56, and progesterone receptor with negativity for the other tested markers.

The size, shape, and architectural distribution of sporadic MMNs did not differ from those observed in DPM. Finally, as previously reported [5], no MMNs were observed in the lung parenchyma of 20 fetal/neonatal autopsies.

Forty-four cases of DPM were collected from the literature showing a female-to-male ratio of 10:1 (40 women and 4 men) with a mean age at diagnosis of 60 years (range 30–80 years) (see Table 2) [8,9,10,11,12,13,14,15,16,17,18,19,20,21,22,23,24,25,26,27,28,29,30,31,32,33,34,35,36,37,38]. 

Nine patients were former or current smokers, and 12 were never-smokers; no data were reported in the remaining cases. Of note, 19 patients (43%) had respiratory symptoms possibly related to DPM, mainly including dry cough and dyspnea, while 25 cases were incidentally discovered. At chest CT, symptomatic and incidental cases had similar findings with randomly-distributed, diffuse, and bilateral micronodules with/without ground-glass opacities (14 cases) and/or central lucency/cavitation (8 cases). MMNs were diagnosed via surgical lung biopsies in 39 cases (88.6%, and surgery was the first choice among invasive procedures in 33 cases) and via transbronchial biopsy in 6 (4 conventional and 2 by cryobiopsy). In five cases, surgical lung biopsy was performed after a non-diagnostic transbronchial biopsy (four cases) or CT-guided fine-needle aspiration cytology (two cases). Only four reported cases had subsequent radiologic investigation for concomitant or metachronous intracranial lesions, which resulted in negative in all four. No specific therapies for DPM were given, and all 44 patients were alive with stable disease at follow-up. Of note, 12 patients had a history of malignancy, and the most common co-morbidity was vascular hypertension in 12 cases (27%). Surprisingly, there were two patients with Turner’s syndrome.

## 4. Discussion

Pulmonary minute meningothelial-like nodules (MMNs) are tiny nests of spindled-to-epithelioid cells generally centered on small pulmonary veins, representing common incidental findings acquired during life in the lung, although their functional significance is still uncertain [1,2,3,4,5,6]. The incidence of multiple MMNs in pulmonary resections ranges from 1.1% to 9.5% [5,6,36]. Some authors have suggested that MMNs may be present in all abnormal lungs with chronic lung disease if sufficiently sampled, particularly in women in their sixth decade of life, in whom hormonal stimulation and chronic lung diseases with hypoxemic conditions possibly induce the growth of interstitial meningothelial cells [5]. Indeed, MMNs have been associated with a variety of pathological conditions, namely, thromboembolism pulmonary malignancies (mainly adenocarcinoma), smoking-related interstitial lung disease, infarcts, and cardiac diseases [1,2,3,5,6,7,8,9,10,11,12,13,14,15,16,17,18,19,20,21,22,23,24,25,26,27,28,29,30,31,32,33,34,35,36,37,38,40]. 

According to previous studies, we further confirm here that MMNs are lacking in fetal and neonatal lung tissue, then supporting their acquired origin [5].

In the great majority of cases, MMNs represent a clinically-irrelevant incidental finding, while their occurrence in the clinic–radiologic setting of interstitial lung disease with micronodular pattern is the prerequisite for a diagnosis of diffuse pulmonary meningotheliomatosis (DPM).

Similar to diffuse idiopathic pulmonary neuroendocrine cell hyperplasia (DIPNECH) [49], DPM represents a distinct clinic–radiologic and pathologic entity rather than solely a histological disease. In other words, the diagnosis of DPM mandatorily requires a close integration of histologic evidence of multiple MMNs in the context of a bilateral micronodular disease radiologically (see Table 3).

Clinically, DPM shows a female predominance (female-to-male ratio around 10:1) with a median age at diagnosis around 60 years [8,9,10,11,12,13,14,15,16,17,18,19,20,21,22,23,24,25,26,27,28,29,30,31,32,33,34,35,36,37,38]. Two-thirds of patients are asymptomatic, and the prognosis is good with a prolonged stable disease without any therapy [8,9,10,11,12,13,14,15,16,17,18,19,20,21,22,23,24,25,26,27,28,29,30,31,32,33,34,35,36,37,38].

In the literature, about 43% had respiratory symptoms (mainly dry cough and dyspnea, which are very common and nonspecific), although it is unclear if these symptoms are secondary to DPM; they could represent some other processes that led clinicians to perform a chest CT scan. At CT imaging, DPM consists of diffuse and bilateral lung involvement by tiny (up to 6 mm), randomly-distributed nodules with/without ground-glass appearance and/or central lucency (ring-like structure or “cheerio sign”) [8,9,10,11,12,13,14,15,16,17,18,19,20,21,22,23,24,25,26,27,28,29,30,31,32,33,34,35,36,37,38]. No lobar predilection has been noted. A similar spectrum of findings has also been observed in cases of metastatic meningioma to the lungs [26,45,46,47,48]. Accordingly, a diagnosis of DPM should include a negative past history of meningioma and the exclusion of a concomitant meningioma by imaging studies (e.g., brain magnetic resonance) since primary DPM and secondary DPM (metastatic meningioma with DPM-like presentation) can be identical in pathology and radiology. 

Indeed, in our study, three out of four cases had previous (two cases) or concomitant evidence of small intracranial meningioma (≤1 cm). Notably, in a clinic-pathologic work of MMNs, Peng et al. [47] originally collected 13 patients, 5 of which showed histologic and radiologic features consistent with DPM. However, among the 13 patients, the authors excluded a patient with intracranial meningioma evidencing a diffuse micronodular pattern on CT images consistent with DPM from this study.

Among the 44 cases of DPM previously reported in the literature, only four (9%) had documented evidence of brain imaging, excluding an intracranial meningioma. Thus, we cannot firmly exclude that some of the cases included in Table 2 might instead represent a bland-looking and indolent example of metastatic meningioma to the lungs.

Based on the aforementioned considerations, in the end, we decided to consider three out of four cases here to be more consistent with pulmonary metastasis from intracranial meningioma showing a DPM-like presentation.

From a pathology viewpoint, a diagnosis of DPM requires a histologic sample since none of the reported cases was diagnosed on cytology. In our series, the diagnosis was obtained by conventional transbronchial biopsy (two cases) and SLB (two cases). In the literature, two cases were first approached by cytology, suggesting a histiocytic disorder [31] and undetermined atypical cells [14], while TBB was attempted in eight cases (two with cryoprobe) leading to a correct diagnosis in four cases (two TBB and two cryobiopsies; among the other four cases, three were non-diagnostic and one was misinterpreted as neuroendocrine cell hyperplasia) [8,9,10,11,12,13,14,15,16,17,18,19,20,21,22,23,24,25,26,27,28,29,30,31,32,33,34,35,36,37,38]. Surgical biopsy was used as the first invasive procedure in 33 cases and the second approach in the other cases after the failure of fine-needle aspiration (two cases) or TBB (four cases) [9,10,11,12,13,14,15,16,17,18,19,20,21,22,23,24,25,26,27,28,29,30,31,32,41,42]. In all cases, MMNs consisted of the discrete proliferation of bland-looking spindled-to-epithelioid cells with uniform nuclei and some pseudoinclusions arranged in a fibrotic stroma. The meningothelial cells immunoreacted with EMA, progesterone receptor, and CD56, whereas cytokeratins, melanocytic, muscle, and vascular markers were negative. Immunohistochemical and ultrastructure analyses of MMNs demonstrated identical characteristics to meningiomas. Interestingly, Masago et al. [50] described a case of primary pulmonary meningioma of 1 cm followed by a CT scan for 2 years, possibly arising in a slowly-growing nodule consistent with a MMNs, then suggested that pulmonary meningiomas may represent a giant form of MMNs.

Ionescu et al. [41] investigated microdissected tissue using 20 polymorphic microsatellite markers targeting 11 genomic regions from 16 cases of MMNs and 10 meningiomas; the authors found that some individual MMNs lacked significant genetic alterations and were more consistent with a reactive origin, 33% of MMNs showed LOH affecting seven genomic loci in DPM, and the highest rate of LOH was detected in meningioma, involving several chromosomal regions (60% in 22q, 44% at 1p and 43% at 14q). The authors suggested that at genetic profiling using LOH analysis, DPM could represent a transition condition between reactive MMNs and meningothelial neoplastic proliferation.

By contrast, Niho et al. [42] analyzed the clonality of 11 multiple MMNs in two women based on an X-chromosome-linked polymorphic marker, the human androgen receptor gene (HUMARA). Six of 11 lesions showed monoclonal expansion. Five lesions in a case with multiple MMNs showed different patterns of monoclonality. These findings showed that some MMNs might show monoclonal expansion, whereas others were polyclonal, possibly indicating a reactive rather than neoplastic significance. Weissferdt et al. [44] explored the genetic setup of the neurofibromatosis (NF) 2 gene on 22q, the commonest genetic abnormality in meningioma, in nine pleuropulmonary meningothelial lesions (six multiple MMNs and three PPMs) and nine intracranial meningiomas by fluorescence in situ hybridization. Deletion of the NF2 gene was identified in two cases with multiple MMNs, one PPM, and four CNS meningiomas. Chromosomal gains of 22q were noted in two cases of MPMN and one PPM. The authors argued that pleuropulmonary meningothelial lesions harbored common genetic alterations with intracranial meningiomas, suggesting that MMNs and meningiomas may arise from the same precursor cell. Accordingly, Higuchi et al. [43] identified deletions of the *NF-2* gene by fluorescence in situ hybridization in two MPMNs and one brain meningioma, suggesting a common pathogenesis through *NF-2* translocation. 

On pathologic ground, the differential diagnosis of primary DPM mainly includes metastatic meningioma, metastatic sarcoma (particularly endometrial stromal sarcoma), DIPNECH, and lymphangioleiomyomatosis (LAM) [51,52,53,54].

Pulmonary metastatic meningioma may show a micronodular pattern, and the diagnosis requires a close correlation with the clinical data since the immunohistochemical profile and even molecular findings between meningioma and MMNs overlap.

Sarcomas of various types metastasize to the lungs, also possibly presenting with miliariform pattern and cystic changes. Knowledge of medical history and the use of specific immunostains (e.g., CD10 and estrogen receptors for endometrial stromal sarcoma) and/or molecular markers (e.g., SYT rearrangement for synovial sarcoma) consistently enable a definitive diagnosis.

Neuroendocrine cell hyperplasia and tumorlets in DIPNECH have a tiny proliferation of spindled-to-oval cells expressing CD56, but they have a bronchiolocentric distribution and express cytokeratins and neuroendocrine markers (e.g., chromogranin, synaptophysin).

LAM is a low-grade neoplasm diffusely affecting the lungs of young women with a nodular and cystic pattern due to irregular interstitial growth of the perivascular epithelioid cell (PEC) system cells, expressing HMB45 and cathepsin K.

Finally, it is noteworthy to mention that pulmonary meningothelial-like nodules seem to consistently express somatostatin receptor 2a (SSTR2a), possibly representing a therapeutic target with specific inhibitors in progressing DPM, should such a case be encountered [55].

## 5. Conclusions

In summary, we collected 4 cases consistent with DPM and reviewed 44 cases previously reported in the literature. Three of our cases had metachronous or concurrent intracranial meningiomas and were finally considered most consistent with a DPM-like pattern of metastatic meningioma to the lungs. Only 9% of DPM in the literature have imaging studies excluding meningioma. Overall, there are three main conditions sustained by the multiple meningothelial cells in the lung, as summarized in Table 4.

We think that DPM should be reserved for a process restricted to the lung in patients in whom there is no evidence of meningioma elsewhere, particularly intracranial. Multiple MMNs is the correct definition in cases showing multifocal MMNs at histology without radiologic changes. Metastatic meningioma with DPM-like features should be used in cases with histologic and radiologic findings mimicking DPM but in the presence of intracranial or extracranial meningioma. Finally, we cannot disprove that occurrence of DPM and meningioma at some other sites could represent a multifocal proliferation of meningothelial cells analogously to what is observed in other diseases with uncertain primary causes (e.g., epithelioid hemangioendothelioma, LAM). Anyway, DPM with/without meningioma seems to pursue an indolent outcome without significant morbidity.

## Figures and Tables

**Figure 1 diagnostics-13-00802-f001:**
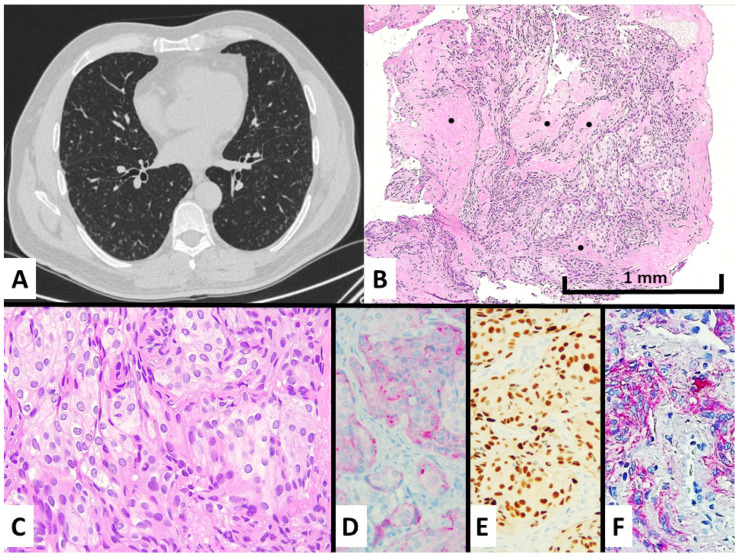
Case #1 manifested with an incidental discovery of diffuse bilateral micronodules also showing ground glass appearance and central cavitation at chest CT scan (**A**) and spindled-to-epithelioid cell proliferation with intermingled fibrosis (see black dots) at transbronchial biopsy (**B**), hematoxylin–eosin magnification ×100, consisting of bland-looking cells with moderate cytoplasm lacking mitotic figures (**C**), hematoxylin–eosin stain magnification ×200. These cells showed a meningothelial cell differentiation by expressing EMA at cytoplasmic level (**D**), immunohistochemistry magnification ×200, progesterone receptors in the nuclei (**E**), immunohistochemistry magnification ×200, CD56 in cytoplasm and membrane (**F**), and immunohistochemistry magnification ×200.

**Figure 2 diagnostics-13-00802-f002:**
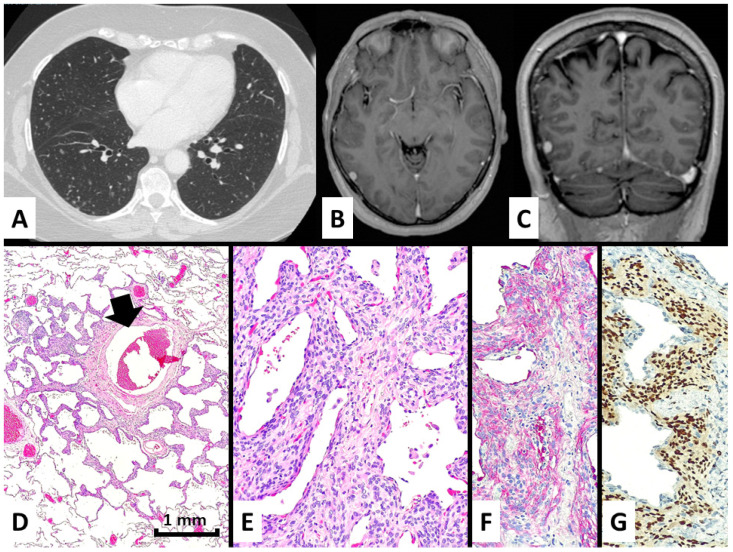
In case #2, the patient presented with several bilateral micronodules with ground glass periphery and some cavitation at chest CT scan (**A**) and had a previous history of non-excised meningioma in the right frontal region (**B**,**C**). A surgical lung biopsy demonstrated a perivenular (black arrow) and irregular proliferation of meningothelial-like cells (**D**), hematoxylin-eosin magnification ×40, growing and thickening the alveolar interstitium (**E**), hematoxylin-eosin stain magnification ×200 and expressing EMA in the cytoplasm (**F**), immunohistochemistry magnification ×200, and progesterone receptors in the nuclei (**G**), immunohistochemistry magnification ×200.

**Table 1 diagnostics-13-00802-t001:** Summary of the main clinical pathologic features of patients with diffuse pulmonary meningothelial nodules in this study.

Case (Year at Diagnosis of DPM)	Gender	Age	Smoke	Symptoms	CT Findings	Sampling Procedure	Original Clinic–Radiologic Diagnosis	Brain Imaging	Specific Therapy	Outcome
#1 (2018)	M	55	N	No; thorax CT performed after car accident	Diffuse bilateral micronodules randomly distributed; some with cavitation	TBB	Sarcoid	Yes; negative	No	Alive and well (38 months)
#2 (2016)	F	56	Y	No; CT performed during a medical check for abdominal discomfort	Diffuse bilateral micronodules randomly distributed	TBB	Sarcoid	Yes; presence of a right frontal lesion (8 mm in maximum diameter) consistent with meningioma (no surgery was performed)	No	Alive and well (63 months)
#3 (2013)	F	68	N	No; incidentally discovered together with a minimally-invasive adenocarcinoma)	Diffuse bilateral micronodules randomly distributed	SLB	Pulmonary metastasis	Yes; previous unknown history of left frontal meningioma (grade 1; 10 mm in maximum diameter)	No	Alive and well (99 months)
#4 (2015)	F	51	N	No; CT scan performed for acute cholecystitis	Diffuse bilateral micronodules with GGO	SLB	DIPNECH	Yes; previous unknown history of right temporal meningioma (4 mm in maximum diameter) without surgery	No	Alive and well (78 months)

Abbreviations: F, female; M, male; Y, current/former smoker; N, never-smoker; SLB, surgical lung biopsy; TBB, transbronchial biopsy; CT-FNA; CT-guided fine needle aspiration; GGO, ground-glass opacities; DIPNECH, diffuse idiopathic pulmonary neuroendocrine cell hyperplasia.

**Table 2 diagnostics-13-00802-t002:** Summary of the main clinical features of diffuse pulmonary meningotheliomatosis with clinic–radiologic details reported in the literature.

Reference	Gender	Age	Smoke	Symptoms	Past Medical History	CT Findings	Sampling Procedure	Brain Imaging	Specific Therapy	Outcome
Suster/Moran [9]	F	75	NA	Dyspnea, shortness of breath, fatigue	Ischemic heart disease; hypertension; colon cancer	Diffuse bilateral reticulonodular infiltrates	SLB	No	No	Alive
Suster/Moran [9]	F	54	NA	Spontaneous pnx, dyspnea	Uterine leiomyosarcoma	Diffuse bilateral reticulonodular infiltrates	SLB	No	No	Alive
Suster/Moran [9]	M	51	NA	Cough, shortness of breath		Diffuse bilateral reticulonodular infiltrates	SLB	No	No	Alive
Suster/Moran [9]	F	63	NA	Cough, shortness of breath	Coronary artery disease; hypertension	Diffuse bilateral reticulonodular infiltrates	SLB	No	No	Alive
Suster/Moran [9]	F	71	NA	Dyspnea, shortness of breath, fatigue	Papillary thyroid carcinoma; invasive breast cancer; lung adenocarcinoma	Diffuse bilateral reticulonodular infiltrates	SLB	No	No	Alive
Jayaschandran et al. [11]	F	74	N	No	Hypertension; dyslipidemia; prediabetes; asthma	Diffuse bilateral micronodules up to 5 mm with GGO and cavitation	TBB	No	No	Alive
Sellami et al. [12]	F	64	Y	Shortness of breath	GERD; hysterectomy; lung adenocarcinoma	Diffuse bilateral micronodules randomly distributed	SLB	No	No	Alive
Kraushaar et al. [13]	F	54	N	No	Hypertension; GERD	Diffuse bilateral micronodules up to 4 mm randomly distributed with thin-walled cavitation	SLB	No	No	Alive
Lee et al. [14]	M	52	Y	Persistent cough and mild dyspnea	Erythrocytosis	Diffuse bilateral micronodules up to 11 mm	CT-guided PNA with atypical cells SLB	No	No	Alive
Sen et al. [15]	F	66	Y	Dry cough	Diabetes mellitus; hypertension; chronic renal disease	Diffuse bilateral micronodules up to 4 mm randomly distributed	SLB	No	No	Alive
Harada et al. [16]	F	56	N	No	Turner’s syndrome; iron deficiency anemia	Diffuse bilateral micronodules up to 5 mm with GGO and central lucency ring-like	SLB	Yes	No	Alive
Maasdorp et al. [17]	F	58	N	Exertional dyspnea, dry cough	Frontal lobe syndrome due to trauma	Diffuse bilateral micronodules with miliary pattern and GGO	SLB	Yes	No	Alive
Alkurashi et al. [18]	F	55	N	Non-exertional dyspnea, dry cough	Prediabetes; hypertension; hypercholesterolemia; hypothyroidism; GERD	Diffuse bilateral micronodules with GGO and cavitation	SLB	No	No	Alive
Dzian et al. [19]	M	60	Y	No	Carotid artery disease with stroke; hypertension; nephro/urolithiasis;	Diffuse bilateral micronodules up to 5 mm	TBB with neuroendocrine cell hyperplasia; SLB	No	No	Alive
Yun et al. [20]	F	80	N	Dry cough, progressive dyspnea	Hypertension; type 2 diabetes mellitus; GERD; chronic kidney disease; colon polyps	Diffuse bilateral micronodules with GGO up to 4 mm	SLB	No	No	Alive
Fernandez Sarabia et al. [21]	F	66	NA	No	Invasive breast cancer; colon adenocarcinoma;	Diffuse bilateral micronodules	SLB	No	No	Alive
Shimaoka et al. [22]	F	54	N	No	Turner’s syndrome	Diffuse bilateral micronodules	SLB	No	No	Alive
Noguchi-Konaka et al. [23]	F	77	NA	No	Breast cancer; colorectal cancer	Diffuse bilateral micronodules	TBB non diagnostic SLB	No	No	Alive
Ding et al. [24]	F	68	NA	Intermittent cough, expectoration		Diffuse bilateral micronodules	SLB	No	No	Alive
Mizutani et al. [25]	F	57	NA	No		Diffuse bilateral micronodules	SLB	No	No	Alive
Park et al. [26]	F	56	NA	No	Breast cancer	Diffuse bilateral micronodules	SLB	No	No	Alive
Bernabeu Mora et al. [27]	F	59	Y	No	Uterine carcinoma; anal abscesses	Diffuse bilateral, randomly-distributed micronodules, most poorly-defined with a cotton wool appearance and cavitation	TBB	No	No	Alive
Gleason et al. [28]	F	63	Y	No		Diffuse bilateral micronodules up to 4 mm with miliary pattern	TBB	No	No	Alive
Morresi-Hauf et al. [29]	F	73	N	No	Goiter	Diffuse bilateral micronodules randomly distributed	SLB	No	No	Alive
Morresi-Hauf et al. [29]	F	60	Y	No	Diabetes mellitus type 2; hypertension; GERD; prurigo nodularis	Diffuse bilateral micronodules randomly distributed	Cryo-TBB	No	No	Alive
Kumar et al. [30]	F	58	N	Intermitted dry cough		Diffuse bilateral micronodules with GGO	TBB not diagnostic SLB	No	No	Alive
Huang et al. [31]	F	57	N	No	Hypercholesterolemia; obesity; hypertension; asthma	Diffuse bilateral micronodules up to 9 mm with GGO	CT-FNA (inflammatory histiocytic process/LCH) SLB	No	No	Alive
Tzilas and Bouros [32]	F	58	N	Chronic cough		Diffuse bilateral micronodules with a peripheral zone predilection and central cavitation (“cheerio sign”)	SLB	No	No	Alive
Swenson et al. [33]	F	61	Y	Dry cough and non-exertional dyspnea		Diffuse bilateral micronodules with an upper lobe predilection	Cryo-TBB	No	No	Alive
Kim et al. [34]	F	55	N	Shortness of breath and exertional dyspnea	Neurodermatitis with prurigo nodularis Obesity and hysterectomy	Innumerable small ground glass nodules, some with cavitation	SLB	No	No	Alive and well
Murata et al. [36]	F	54	Y	No	Cholelithiasis; hypercholesterolemia	Diffuse bilateral micronodules with a peripheral zone predilection. Some nodules with central lucency and ring-shaped appearance (“cheerio sign”)	TBB non diagnostic SLB	No	No	Alive and well
Wang et al. [37]	F	30	NA	No	Microinvasive lung adenocarcinoma	Multiple GGO nodules	SLB	No	No	Alive and well
Wang et al. [37]	F	40	NA	No	Multiple invasive lung adenocarcinoma	Multiple microscopic nodules	SLB	No	No	Alive and well
Wang et al. [37]	F	50	NA	No		Multiple GGO nodules	SLB	No	No	Alive and well
Wang et al. [37]	F	50	NA	No		Multiple GGO nodules	SLB	No	No	Alive and well
Wang et al. [37]	F	50	NA	Chest pain radiating to the right shoulder		Multiple microscopic nodules	SLB	No	No	Alive and well
Wang et al. [37]	F	56	NA	Cough with yellow sputum	Organizing pneumonia; atypical adenomatous nodules	Multiple microscopic nodules	SLB	No	No	
Wang et al. [37]	M	66	NA	Recurrent cough, shortness of breath, hemoptysis	Nonkeratinizing squamous cell carcinoma; organizing pneumonia	Multiple microscopic nodules	SLB	No	No	NA
Huang et al. [38]	F	64	N	No	Hypertension	Multiple GGO nodules	SLB	Yes (ischemic foci in the left centrum semiovale and mild demyelination of white matter; no intracranial meningioma)	No	NA
Peng et al. [47]	F	63	NA	No	Squamous cell carcinoma of the esophagus	Diffuse bilateral micronodules	SLB	No	No	Alive
Peng et al. [47]	F	68	NA	No		Diffuse bilateral micronodules with GGO	SLB	No	No	Alive
Peng et al. [47]	F	54	NA	No		Diffuse bilateral micronodules with GGO	SLB	No	No	Alive
Peng et al. [47]	F	59	NA	No		Diffuse bilateral micronodules	SLB	No	No	Alive
Asakawa et al. [48]	F	76	NA	NA	Basedow goiter; hypertension	Diffuse bilateral micronodules up to 6 mm	SLB	Yes	No	Alive

Abbreviations: F, female; M, male; Y, current/former smoker; N, never-smoker; NA, not available; SLB, surgical lung biopsy; TBB, transbronchial biopsy; CT-FNA; CT-guided fine needle aspiration; Cryo-TBB, transbronchial biopsy with cryoprobe; GGO, ground-glass opacities; GERD, gastroesophageal reflux disease.

**Table 3 diagnostics-13-00802-t003:** Clinicopathologic characteristics of diffuse pulmonary meningotheliomatosis (DPM).

**Clinical features**
Female > Male (up to 15:1)
Sixth decade
Asymptomatic/incidental presentation/mild nonspecific symptoms
Stable disease and good prognosis
**Radiologic features**
Diffuse, bilateral, and randomly-distributed micronodules up to 6 mm of maximum diameter with/without ground-glass opacities and with/without central “cavitation” (ring-like lucency)
**Pathologic features**
Histologic evidence of MMNs; absence of alternative histologic cause of radiologic nodules
Lack of intra/extracranial meningioma *

Abbreviations: MMNs, minute meningothelial-like nodules; TBB, transbronchial biopsy; SLB, surgical lung biopsy. * If a meningioma is identified previously, concurrently, or subsequently, the possibility/likelihood of metastatic meningioma mimicking DPM has to be considered.

**Table 4 diagnostics-13-00802-t004:** Proposed definitions of conditions with multiple meningothelial-like nodules in the lung.

Definition	Incidence	Histology	Radiology	Other
Diffuse pulmonary meningotheliomatosis (DPM)	Rare; female prevalence	Multiple meningothelial-like nodules with perivenular and interstitial distribution	Diffuse bilateral micronodules (up to 5–6 mm) with a peripheral zone predilection; some nodules show central lucency (“cheerio sign”)	Brain imaging should be performed to exclude intracranial meningioma
Metastatic meningioma with DPM features	Rare; no gender prevalence	Multiple meningothelial-like nodules with perivenular and interstitial distribution	Diffuse bilateral micronodules (up to 5–6 mm) with a peripheral zone predilection; some nodules show central lucency (“cheerio sign”)	Positive past medical history of meningioma or concomitant meningioma
Multiple meningothelial-like nodules	Not infrequent; no gender prevalence	Multiple meningothelial-like nodules with perivenular and interstitial distribution	No changes	Incidental findings in several conditions (infections; primary/metastatic lung cancer; pulmonary infarcts; hypoxemic conditions)

## Data Availability

The data that support the findings of this study are available from the corresponding author upon reasonable request.

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
