# Peer review of "Diffuse Pulmonary Meningotheliomatosis: Clinic-Pathologic Entity or Indolent Metastasis from Meningioma (or Both)?"

_diagnostics, 2023, doi:10.3390/diagnostics13040802_

Round 1

Reviewer 1 Report

Drs. Melocchi et al. report on a small series of cases of diffuse pulmonary meningotheliomatosis and discuss its relationship to meningioma. The manuscript is well-written and cases are presented thoroughly. Aside from a few typographical corrections, I do not see a need for any major modifications. The images, tables and references are adequate and appropriate.

Author Response

We thank the reviewer for the encouraging comment on this manuscript.

Reviewer 2 Report

I read with great interest the present manuscript. It is well-written and includes all relevant literature. The results and the discussion implicates significant criteria to perform further diagnosis or not. In my opinion, the manuscript can be accepted for publication.

Author Response

We thank the reviewer for the generous comment on this study.

Reviewer 3 Report

Manuscript Number: 2224910

The manuscript entitled “Diffuse pulmonary meningotheliomatosis: clinic-pathologic entity or indolent metastasis from meningioma (or both)?    is based upon a very good premise and the results are very promising. The results appear to be valid and the methodology is appropriate. This study report is very interesting and suitable for this journal.

Comments

1. In the Abstract, you can delete the sentence “DPM is an extremely rare occurrence generally discovered in asymptomatic 29 adult patients with a female predominance” (lane 29).

2. Lane 41 in abstract, “Diagnostic criteria for primary 41 DPM are proposed…. also can delete

3. Here mentioned about 4 cases of DPM from the archival files were taken for this study, have you had any criteria for that? The patient's gender, background, etc,

4. In the result section, Table 1 is not appropriately depicted. Please check and correct it.

5. The cellular growth was often accompanied by variable 135 amounts of collagenized fibrous tissue. Did you use any collagen detection staining in this study?

6. In figure 1, please put a scale bar for each image from A to F, as well size of the image made similarly.

7. In histopathological analysis, if you can differentiate any change in the image please mark it accordingly.

8. How do you recognize the epithelial mem-148 brane antigen (EMA), progesterone receptor and CD56 in immunohistochemical staining?

9. Figure 2, insert scale bar in the picture

10. Significance of the progesterone receptor in this condition? Is the progesterone receptors help in cell proliferation?

11. Table 2 also please modify and represent in a good format.

The manuscript is recommended for acceptance after this minor modification.

Author Response

Authors’reply to the Reviewer’s comments

Manuscript Number: DIAGNOSTICS 2224910

Reviewer #1

The manuscript entitled “Diffuse pulmonary meningotheliomatosis: clinic-pathologic entity or indolent metastasis from meningioma (or both)?”  is based upon a very good premise and the results are very promising. The results appear to be valid and the methodology is appropriate. This study report is very interesting and suitable for this journal.

Comments

  1. In the Abstract, you can delete the sentence “DPM is an extremely rare occurrence generally discovered in asymptomatic 29 adult patients with a female predominance” (lane 29).

Authors’reply. According to the reviewer’s comment, we delete the aforementioned sentence.

  1. Lane 41 in abstract, “Diagnostic criteria for primary 41 DPM are proposed…. also, can delete

Authors’reply. According to the reviewer’s comment, we delete the aforementioned sentence.

  1. Here mentioned about 4 cases of DPM from the archival files were taken for this study, have you had any criteria for that? The patient's gender, background, etc,

Authors’reply. The reviewer’s point is interesting and we added a brief sentence on case selection. Nevertheless, the case series merely represent incidental and consecutive cases identified during the routine practice or the consultation activity (the revised text is in red capitals). Unfortunately, there are no clinic-radiologic characteristics that may specifically select patients with DPM.

  1. In the result section, Table 1 is not appropriately depicted. Please check and correct it.

Authors’reply. In agreement with the reviewer’s suggestion, we added more details in the Results section of the paper concerning our case series (in red capitals in the revised version).

  1. The cellular growth was often accompanied by variable amounts of collagenized fibrous tissue. Did you use any collagen detection staining in this study?

Authors’reply. We did not used staining for collagen since H&E stain is generally sufficiently diagnostic in recognition of collagen tissue. Collagen fibrous tissue is frequently seen in the meningothelial proliferations of the lung.

  1. In figure 1, please put a scale bar for each image from A to F, as well size of the image made similarly.

Authors’reply. According to the reviewer’s comment, we added a scale bar in the low magnification image, since the size of the biopsy may have some interest for the readers. Magnification x the objective used to catch the other images is sufficiently informative, also in agreement with the Journal style.

  1. In histopathological analysis, if you can differentiate any change in the image, please mark it accordingly.

Authors’reply. According to the reviewer’s comment, we added some marks (black dot indicating the fibrosis in Figure 1 and black arrow indicating the pulmonary venule in Figure 2). The legend of both Figures was modified accordingly (in red capitals).

  1. How do you recognize the epithelial membrane antigen (EMA), progesterone receptor and CD56 in immunohistochemical staining?

Authors’reply. In agreement with the reviewer’s comment, we added more details in the Figures legend (red capitals) indicating that EMA has a cytoplasmic stain, PgR is expressed in the nuclei and CD56 is characterized by a cytoplasmic and membranous reactivity.

  1. Figure 2, insert scale bar in the picture

Authors’reply. According to the reviewer’s comment, we added a scale bar in the low magnification image, since the size of the biopsy may have some interest for the readers. Magnification x the objective used to catch the other images is sufficiently informative, also in agreement with the Journal style.

  1. Significance of the progesterone receptor in this condition? Is the progesterone receptors help in cell proliferation?

Authors’reply. Progesterone receptor in meningothelial proliferation has a diagnostic rather than a cell proliferation role. Some authors have suggested a predictive value in treating meningothelial proliferation with hormonal therapy, but no efficacy was recorded using these treatments.

  1. Table 2 also please modify and represent in a good format.

Authors’reply. We agree with the reviewer’s comment, but the format of the tables is fixed by the Editing Board of the Journal. We’ll try to improve the format of the tables once received the proofs of the manuscript, if possible.

The manuscript is recommended for acceptance after this minor modification.

All the authors thank the reviewer for the suggested comments and changes, aimed at improving the quality and the interest of the study.